# The Pivotal Role of Protein Phosphatase 2A (PP2A) in Brain Tumors

**DOI:** 10.3390/ijms232415717

**Published:** 2022-12-11

**Authors:** Laura Cucinotta, Alessia Filippone, Giovanna Casili, Marika Lanza, Valentina Bova, Anna Paola Capra, Raffaella Giuffrida, Cristina Colarossi, Dorotea Sciacca, Irene Paterniti, Salvatore Cuzzocrea, Michela Campolo, Emanuela Esposito

**Affiliations:** 1Department of Chemical, Biological, Pharmaceutical and Environmental Sciences, University of Messina, Viale Ferdinando Stagno D’Alcontres, 31-98166 Messina, Italy; 2Istituto Oncologico del Mediterraneo, Via Penninazzo 7, 95029 Viagrande, Italy

**Keywords:** PP2A, PP2A inhibitors, cancer, brain tumors, GBM, CIP2A inhibition

## Abstract

Protein phosphatase 2A (PP2A) is a highly complex heterotrimeric Ser/Thr phosphatase that regulates many cellular processes. PP2A is dysregulated in several human diseases, including oncological pathology; interestingly, PP2A appears to be essential for controlling cell growth and may be involved in cancer development. The role of PP2A as a tumor suppressor has been extensively studied and reviewed. To leverage the potential clinical utility of combination PP2A inhibition and radiotherapy treatment, it is vital that novel highly specific PP2A inhibitors be developed. In this review, the existing literature on the role of PP2A in brain tumors, especially in gliomas and glioblastoma (GBM), was analyzed. Interestingly, the review focused on the role of PP2A inhibitors, focusing on CIP2A inhibition, as CIP2A participated in tumor cell growth by stimulating cell-renewal survival, cellular proliferation, evasion of senescence and inhibition of apoptosis. This review suggested CIP2A inhibition as a promising strategy in oncology target therapy.

## 1. Introduction

Reversible protein phosphorylation by kinases and phosphatases is a crucial mechanism of cancer cell signal transduction. Most protein phosphorylation occurs at serines and threonines (Ser/Thr), and the protein phosphatase 2A (PP2A) appears to contribute significantly to Ser/Thr dephosphorylation activity in human cells. PP2A is a trimeric protein complex consisting of a central dimer formed between a scaffolding A-subunit (PPP2R1A and PPP2R1B) and a catalytic C-subunit (PPP2CA and PPP2CB), which is associated with one of the many B-subunits that facilitate the interaction of the trimer with substrate proteins. In addition to its importance in various physiological processes, PP2A is also an important human tumor suppressor, whose inhibition promotes malignant transformation of normal cells [1]. Recent studies have shown that PP2A inhibits numerous growth and survival pathways, suggesting that the ability to activate PP2A in cancer may suppress the development of resistance, providing greater efficacy in reducing tumor burden when combined with impediments specific to pro-proliferative pathways [2]. It has been shown that in more than 50% (59.6%) of glioblastoma cases, the activity of PP2A is inactivated or dysregulated due to overexpression of its inhibitors, especially SET and CIP2A. This review analyzes existing literature on the role of PP2A in tumors, mainly in gliomas and glioblastoma (GBM). In particular, this review is focused on the role of CIP2A, an inhibitor of PP2A, which through stimulating cell proliferation and survival, evasion of senescence, and apoptosis inhibition, increases tumor cell growth. This review suggested CIP2A inhibition as a promising strategy in oncology target therapy.

## 2. Role of PP2A in Cancer

Protein phosphatase 2A (PP2A) is a member of the serine/threonine phosphatase family that is dysregulated in different human diseases, such as Alzheimer’s and cardiovascular disease (Figure 1) [3]. PP2A is a critical human tumor suppressor, the inhibition of which is a prerequisite for malignant transformation of many types of normal human cells and promotes in vivo tumorigenesis. PP2A is an important negative regulator of several oncogenic signaling pathways and, particularly, of RAS-driven oncogenic signaling [4]. In addition, PP2A scaffold and regulatory subunits have been found to be mutated or aberrantly expressed in many different types of cancer [5]. Other studies have showed that inhibition of PP2A causes selective lethality to PLK1-overexpressing breast, pancreatic, ovarian, glioblastoma and prostate cancer cells [6]. In early genetic models of malignant transformation, PP2A was described as a tumor suppressor, demonstrating that inhibition of PP2A was critical for the initiation of carcinogenesis. PP2A is now known to inhibit multitudinous growth and survival pathways, suggesting that the ability to activate PP2A in cancer may suppress the development of resistance, providing greater efficacy in reducing tumor burden when combined with specific inhibitors to pro-proliferative pathways [2]. Other studies have showed that alterations in PP2A-mediated signaling pathways cause cancer. The inhibition/inactivation of PP2A could induce apoptotic cell death in various cancer cell models including those of the pancreas, liver, blood and testicles [7]. The inhibition of activity or loss of some of the functional subunits of PP2A is characteristic neoplastic transformation: an alteration of a PP2A subunit or the loss of its phosphatase activity has been linked to the development of cancer. Regarding the mechanism of action of PP2A, it appears that it works by regulating the cell cycle and apoptosis. In fact, PP2A possesses pro-apoptotic activity, being able to negatively regulate the PI3K/Akt pathway after direct dephosphorylation of Akt, inactivate the antiapoptotic Bcl-2, and activate the proapoptotic factor BAD. In particular, direct inactivation of PP2A-dependent Akt and BAD dephosphorylation result in its activation and translocation in the mitochondrial membrane where it binds and inhibits Bcl-2 [8,9]. Recently, cytostatin, a potent anti-metastatic drug, has been shown to be able to selectively inhibit PP2A. In addition, the PR65α and PR65β subunits of PP2A have been identified as tumor suppressors; in fact, their genes are mutated in melanomas and lung and breast cancers for PR65α and in 15% of cell lines derived from the primary lung and colon for PR65β [10]. PP2A inhibits nuclear telomerase activity in human breast cancer cells. Particularly, in normal somatic cells, telomerase activity is so minimal that it is not detected, but in the case of primary human malignant tumors, it turns out to be high, suggesting that the synthesis of telomeres is crucial for unlimited cell division. By inhibiting this increased telomerase activity in cancer cells, it can, therefore, counteract uncontrolled cell growth [11]. The complexity of the structure, function and regulation of PP2A show that the role of PP2A as a tumor suppressor may depend on several factors, such as the cellular context and the subunit involved. In addition, the loss of a specific PP2A activity may contribute to the development and progression of cancer, but also represent a prognostic factor due to the key role played by PP2A as a mediator of antiproliferative and pro-apoptotic signals [8]. Several studies have reported that both the α as well as the β isoforms of the A subunit were genetically altered in a variety of primary human tumors [12,13]. One of these studies [14] described that the gene encoding the β isoform of the A subunit (Aβ) was found to be altered in 6% of lung tumor-derived cell lines, in 15% of primary lung tumors, and in 15% of colorectal carcinomas. Various studies on mutations of the PP2A subunit in human tumors have not guaranteed the tumor suppressor role of this enzyme. Campbell and Manolits [15], in their study, found that in primary ovarian tumors, the mutations detected in the Aβ gene did not differ from mutations present in non-tumor controls. Moreover, since at least one of the mutations represents a non-pathological polymorphism, other reported mutations of the A subunit may not have pathological relevance. Therefore, they concluded that some of the reported mutations of subunit A may not even have any pathological relevance.

Its potential tumor suppressive capacity may be due to the ability of specific regulatory B subunits to direct the holoenzyme to specific positions within the cell and to modulate its activity in the respective subcellular area. An example of this could be the absence of cancer-associated mutations in subunit A. The tumor suppressive function exerted by some B subunits could be inactivated through mutations against them [16]. 

PP2A appears to be critically involved in cell growth control and, potentially, in cancer development. As shown in Figure 2, PP2A activity is indispensable for every cell and takes part in most cellular pathways: PP2A appears to possess a key role in the Wnt signaling pathway and pro-apoptotic activity, being able to downregulate the PI3K/Akt pathway after direct dephosphorylation of Akt to inactivate the antiapoptotic Bcl-2 and to activate the proapoptotic factor.

Although some studies have indicated that this enzyme could exert the suppressive function of the tumor, other studies have shown the importance of PP2A in cell growth and survival [17,18,19]. This is because PP2A is not a single enzyme, but a multitasking enzyme system, existing in multiple isoforms. Therefore, PP2A can exert inhibitory effects on cell growth, as well as stimulators, thanks to the activity of several complexes with distinct and specific subcellular localization [16].

## 3. Role of PP2A in GBM

GBM represents the most malignant and aggressive form of gliomas (defined by the World Health Organization as grade IV), characterized by necrosis, invasiveness and excessive proliferation [20,21]. Despite the research made over the past years, the therapeutics results of GBM still remain poor, since complete resection of tumor mass is difficult due to its high invasiveness in the adjacent brain tissue and the resistance of the residual tumor to chemoradiotherapy. Therefore, new adjuvant therapies for GBM are urgently needed [22]. PP2A is the most studied protein phosphatase in the context of GBM. Several databases (Human Protein Atlas and TCGA) show that in 59.6% of GBM cases, PP2A activity is inactivated or dysregulated, especially due to overexpression of its two inhibitors SET and CIP2A. Recent studies demonstrated that in GBM, inhibition of PP2A activity exerted anti-oncogenic effects. 

Cells subjected to stressful conditions, such as irradiation treatment or hypoxia, often upregulate PP2A expression. These conditions are found in GBM and PP2A and may promote GBM survival by reducing metabolic demand in hypoxia and ATP consumption [23].

Hofstetter et al., through the study of PP2A expression and activity in GBM, investigated whether PP2A activity in GBM is induced by hypoxia and whether it is involved in regulation of cell cycle progression and survival of severely hypoxic tumor cells; interestingly, PP2A mediates the energy consumption reduction of hypoxic TSCs (tumor stem-like cells), which enhances tumor cell survival [24]. Many tumors, such as malignant gliomas, as a result of PP2A inhibition, show slower growth and increased death of apoptotic cells. In malignant gliomas, the inhibition of PP2A increases the frequency of cells in the M phase of mitosis, causing an inhibition of tumor proliferation [25]. PP2A is not genetically inactivated in GBM [26]. Among the mechanisms suggested to induce non-genetic dysregulation of PP2A in GBM is the overactivation of the receptor tyrosine kinases (RTKs), such as the epidermal growth factor receptor (EGFR), by genetic alteration, frequently viewed in GBMs [22]. The functional importance of PP2A as a tumor suppressor in GBM has not yet been sufficiently studied [27]. However, this evidence suggests the potential of PP2A as a therapeutic target for GBM through suppression but also through upregulation of its activity or expression.

## 4. PP2A Inhibitors in GBM

Dysregulation of PP2A is commonly observed in various cancers. Several studies highlighted the tumor suppressive ability of PP2A. Consistent with these findings, PP2A is functionally inactivated in many cancers, typically through one of several different mechanisms such as somatic mutation, loss of heterozygosity and/or decreased expression of PP2A subunits, increased expression of endogenous inhibitors of PP2A and changes in phosphorylation/methylation of the C subunit. Two endogenous inhibitors, I1PP2A and I2PP2A, were first identified as PP2A inhibitors; particularly, I2PP2A, later identified as SET, was found to play a role in myeloid leukemogenesis and acute undifferentiated leukemia [28].

An increasing number of works in the literature have demonstrated that PP2A inhibitors’ antagonization has anti-tumor effects both in vitro and in vivo. The discovery of small-molecule antagonists of PP2A inhibitors, especially CIP2A and SET, has provided insight into the mechanisms of PP2A dysregulation in brain tumors, highlighting a therapeutic avenue with promising potential (Table 1) [25,29]. 

LB100LB100 is a molecule derived from the synthetic anticancer compound norcantharidin [30,31]. LB100 competitively inhibits PP2A by directly binding to PP2A-C and reducing its catalytic activity [32]. Combination treatment with LB100 and radiation significantly delayed tumor growth, prolonging survival [33]. Cui et al. showed that LB100 enhanced the anti-CAIX CAR-T cell cytotoxic activity against GBM tumor cells [34]. Lu et al. demonstrated that LB100 inhibited PP2A and caused dose-dependent antiproliferative activity in GBM cell lines [35]. In another study conducted by the same group, Lu et al. studied LB-102, a lipid-soluble homolog of LB100, demonstrating that it combined with temozolomide, a DNA-methylating chemotherapeutic drug used to treat glioblastoma multiforme, causes complete regression of glioblastoma multiforme (GBM) xenografts. In addition, combined with doxorubicin (DOX), a DNA-intercalating agent used as an anticancer drug, LB-102 causes marked GBM xenograft regression, whereas DOX alone only slows growth. These findings indicate that inhibition of PP2A by LB-102 blocks cell-cycle arrest and increases progression of the cell cycle in the presence of TMZ or DOX [36].

PME-1Protein phosphatase methylesterase 1 (PME-1) is an endogenous PP2A inhibitor protein that regulates PP2Ac activity by demethylating the highly conserved carboxyl-terminal tail and binding directly to the PP2Ac catalytic site. Kaur et al. demonstrated that overexpression of the PP2A inhibitor protein PME-1 drives glioma cell resistance to various multikinase inhibitors through specific PP2A complexes and a decrease in histone deacetylase 4 cytoplasmic activity. PME-1 and HDAC4 are associated with the progression of human glioma [27].

SETOne of the endogenous inhibitors of PP2A is SET nuclear proto-oncogene (SET) [37]. In the context of GBM, numerous sources of evidence suggest that SET could represent a potential carcinogenic factor. Through the upregulation of Bcl-2 gene expression and downregulation of Bax and caspase-3 expression, it is able to regulate cell proliferation and apoptosis of GBM cells [38].

CIP2AAnother endogenous inhibitor of PP2A is the cancerous inhibitor of PP2A (CIP2A), an oncoprotein upregulated in several peripheral tumors. CIP2A is able to promote the growth of cancer cells through the inhibition of dephosphorylation of PP2A substrates involved in cancer development. Moreover, CIP2A is found to be upregulated in mouse brain astrocytes (causing reactive astrogliosis), which promote synaptic degeneration and cognitive deficits [29]. CIP2A, directly interacting with the oncogenic transcription factor c-Myc, inhibits the activity of PP2A toward c-Myc serine 62 (S62) and, thus, prevents c-Myc proteolytic degradation. In addition, CIP2A promotes in vivo tumor formation and anchorage-independent cell growth [29].

## 5. CIP2A

CIP2A is an oncoprotein encoded by the KIAA1524 gene [39], expressed mainly in the cytoplasm of cancerous cells and upregulated in most human malignancies [40,41]. In addition, recent studies showed that CIP2A is an endogenous inhibitor of PP2A in malignant cells, thereby promoting the transformation of defective cells into malignancy and tumor growth [42]. CIP2A overexpression enhances invasiveness and metastatic behavior of cancers and also acts as a useful marker in diagnosis, treatment and prognosis of these cancers [43]. As a result, inhibition of CIP2A could be used as a promising therapeutic target in cancer therapy [44,45]. Recent studies demonstrated an overexpression of CIP2A in several cancer cells [41], such as tumors of the stomach [46], esophagus [47], colon [48] and pancreas [49], renal cell cancer [50] and ovarian and cervical cancer [51,52]. The mechanism of PP2A inhibition by CIP2A is not well-understood, although some suggest that CIP2A binds allosterically to PP2A’s binding interface of the B subunit, thus altering substrate specificity and/or limiting activity. Recent studies have shown that silencing of CIP2A by small interfering RNAs (siRNA) inhibits the growth of xenografted tumors of various types of cancer cells [29]. The activation and overexpression of CIP2A in tumor cells have been investigated by several studies in which CIP2A was found to be associated with several molecules of the signaling pathways that regulate tumor proliferation and apoptosis, including C-Myc, DAPK1 (death-associated protein kinase 1), E2F1 (oncogenic transcription factor) and Plk1 (polo-like kinase 1) [53,54]. CIP2A works by preventing PP2A-mediated dephosphorylation of the c-Myc oncoprotein to serine 62, leading to increased proliferation and progenitor cell renewal. CIP2A-mediated PP2A inhibition results in stabilization of the Myc protein, resulting in increased availability of this oncoprotein [55]. Furthermore, Kim et al. demonstrated the involvement of CIP2A in the regulation of the cell cycle and in mitosis, identifying Plk1 as a target of CIP2A [56]. In addition, CIP2A associates with mTORC1 (mammalian target of rapamycin complex 1), acting as an allosteric inhibitor of mTORC1-associated PP2A, thereby enhancing mTORC1-dependent growth signaling and inhibiting autophagy [57]. PP2A acts as a negative regulator of MEK and ERK; consequently, the inhibition of PP2A by CIP2A causes the activation of these molecules. In summary, CIP2A activates E2F1 and Plk1 and inhibits DAPK1, MEK and ERK. Thus, CIP2A is involved in tumor cell growth by stimulating cell proliferation, survival of cell renewal, evasion of senescence and inhibition of apoptosis (Figure 3) [53].

## 6. Role of CIP2A in Glioma and GBM

The oncoprotein CIP2A is overexpressed in several types of tumors and promotes the proliferation and transformation of cancer cells. However, its function in glioma is poorly understood. Therefore, further studies are needed to understand whether CIP2A may represent a new drug target for glioma. In gliomas, CIP2A is overexpressed and associated with tumor size, WHO grade and overall postoperative survival rate. Therefore, depletion of CIP2A inhibits the cellular proliferation of gliomas [58]. Xu et al. found that the expression of CIP2A was higher in cancerous glioma tissues than that in normal tissues; in particular, higher levels of expression were found in high-grade gliomas (grade III–IV) than in low-grade gliomas (grade I–II). Their study also demonstrated that CIP2A depletion inhibited glioma cell proliferation, migration and invasion. Therefore, they hypothesize that CIP2A was involved in glioma progression, indicating that CIP2A could be used as a potential therapeutic target in the future [59]. Similarly, Khanna et al., in their study, showed a decrease in expression for all PP2A subunits in glioma tissues compared to healthy tissues, further strengthening the evidence of the oncogenic role of CIP2A in gliomas. Depletion of CIP2A expression in GBM cells induces senescence and prevents tumor growth in vitro and in vivo [60]. Gao et al. demonstrated that knockdown of CIP2A in GBM cell lines LN229 and A172 led to an inhibition of cell activity and cell cycle arrest [43]. Qin et al., in their study, demonstrated that the overexpression of CIP2A promotes invasive behavior in GBM, and a natural compound, cucurbitacin B (CuB), shows an anti-proliferative and anti-invasion effect in GBM cell lines [61]. These findings represent a basis for the development of alternative strategies for the clinical treatment of glioma. In summary, CIP2A is involved in the development and progression of glioma, but its mechanisms in gliomas are still complex and unclear, so further studies are needed to define its mechanism of action.

## 7. Conclusions and Future Directions

Despite positive results and numerous ongoing clinical trials, there are limitations to the development of these new candidates due to the pharmacokinetic complexity. Overall, PP2A represents an engaging drug target in cancer, particularly in brain tumors, and the complexity of PP2A regulation emphasizes the need for further refinement and approaches. Therefore, considering the promising effect of PP2A inhibitors, they could represent a valid target therapy for brain cancer.

## Figures and Tables

**Figure 1 ijms-23-15717-f001:**
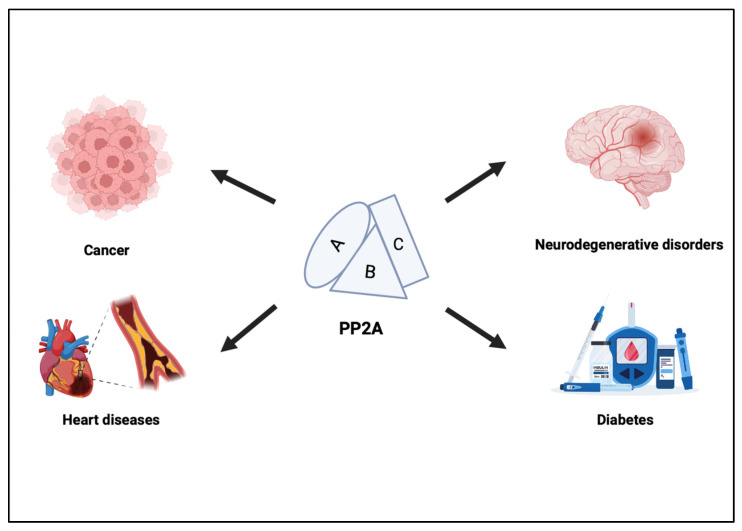
The physiological role of PP2A.

**Figure 2 ijms-23-15717-f002:**
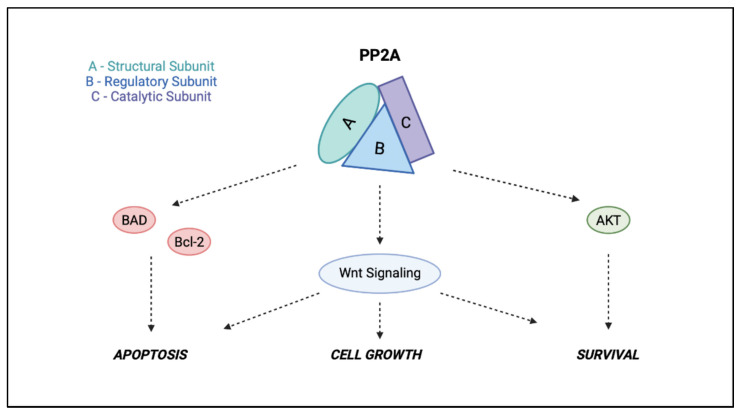
The involvement of PP2A in cancer.

**Figure 3 ijms-23-15717-f003:**
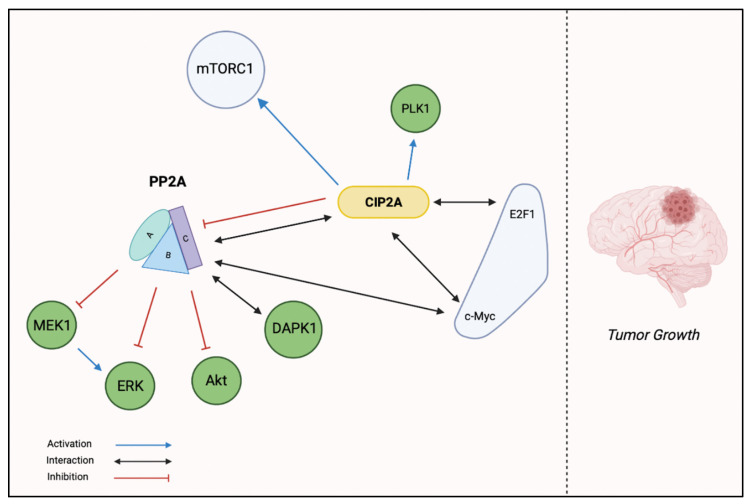
The involvement of CIP2A in tumor growth.

**Table 1 ijms-23-15717-t001:** PP2A inhibitors in glioblastoma.

Inhibitor	Mode of Inhibition	References
LB100	Binds to PP2A-C and reduces its catalytic activity	[30,31,32,33,34,35,36]
PME-1	Demethylation of L309 on the subunit C of PP2A	[27]
SET	Binds to and inactivates PP2A	[37,38]
CIP2A	Binds to PP2A, preventing the dephosphorylation of c-MYC and Akt	[29]

## Data Availability

Not applicable.

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
