# Peer review of "The Pivotal Role of Protein Phosphatase 2A (PP2A) in Brain Tumors"

_ijms, 2022, doi:10.3390/ijms232415717_

Round 1
Reviewer 1 Report
In this review, Laura Cucinotta et al. analyzed the existing literature on the role of PP2A in brain tumors, especially in glioma and glioblastoma (GBM). Interestingly, the review focused on the role of PP2A inhibitors, focusing on CIP2A inhibitor, as CIP2A partecipated in tumor cell growth by stimulating cell – renewal survival, cellular proliferation, evasion of senescence and inhibition of apoptosis. This review suggested CIP2A inhibition as a promising strategy in oncology target-therapy. Their work made people more clearly understand the important role of PP2A in brain tumors, and provided new ideas for the treatment of brain tumors, especially glioma. Having said that, the following issues have to be addressed before it can be published.
1.Line 60-62, “Cunningham et al. found that inhibition of PP2A causes selective lethality to PLK1-overexpressing breast, pancreatic, ovarian, glioblastoma, and prostate cancer cells” indicates PP2A's tumor-promoting rather than suppressive ability, contrary to the general meaning of this part, and should be placed after ”Other studies have showed that…”.
2. In my opinion, the content of Figure 2 should be described in the text. There should be a description below each picture in the passage.
Author Response
In this review, Laura Cucinotta et al. analyzed the existing literature on the role of PP2A in brain tumors, especially in glioma and glioblastoma (GBM). Interestingly, the review focused on the role of PP2A inhibitors, focusing on CIP2A inhibitor, as CIP2A partecipated in tumor cell growth by stimulating cell – renewal survival, cellular proliferation, evasion of senescence and inhibition of apoptosis. This review suggested CIP2A inhibition as a promising strategy in oncology target-therapy. Their work made people more clearly understand the important role of PP2A in brain tumors and provided new ideas for the treatment of brain tumors, especially glioma. Having said that, the following issues have to be addressed before it can be published.
1.Line 60-62, “Cunningham et al. found that inhibition of PP2A causes selective lethality to PLK1-overexpressing breast, pancreatic, ovarian, glioblastoma, and prostate cancer cells” indicates PP2A's tumor-promoting rather than suppressive ability, contrary to the general meaning of this part, and should be placed after ”Other studies have showed that…”.
In according with the reviewer, the authors have adjusted this sentence (page 2, line 59).
- In my opinion, the content of Figure 2 should be described in the text.There should be a description below each picture in the passage.
As suggested by the reviewer, the authors better described the Figure 2 in the text (page 3, line 111).
Thank you for the comments.
Reviewer 2 Report
This manuscript reviewed the research progress of PP2A in cancer and mainly focused on the potential clinical significance in brain tumors. It is well-organized and reasonably written.
Some minor concerns: 1, The resolution of the figures is poor. The authors had better improving it.
2, some grammatical errors: Such as: numerous evidences (evidence) suggest (suggests) that SET could represent a potential carcinogenic factor, as through upregulation of Bcl‐2 gene expression, and downregulation of Bax and caspase‐3 expression, it is able to regulates (regulate) cell proliferation and apoptosis of GBM cells.
3, Page 5 line 188: Doxorubicin (Dox): abbreviated firstly, then it can be used other places.
Author Response
This manuscript reviewed the research progress of PP2A in cancer and mainly focused on the potential clinical significance in brain tumors. It is well-organized and reasonably written.
Some minor concerns:
1, The resolution of the figures is poor. The authors had better improving it.
In according with the reviewer, the authors increased the resolution of the figures.
2, some grammatical errors: Such as: numerous evidences (evidence) suggest (suggests) that SET could represent a potential carcinogenic factor, as through upregulation of Bcl‐2 gene expression, and downregulation of Bax and caspase‐3 expression, it is able to regulates (regulate) cell proliferation and apoptosis of GBM cells.
As suggested by the reviewer, the authors corrected the grammatical errors.
3, Page 5 line 188: Doxorubicin (Dox): abbreviated firstly, then it can be used other places.
As suggested by the reviewer, the authors added the abbreviation first (DOX) at line 191.
Thank you for the comments.